# Using the Axial Oblique View of Computed Tomography (CT) in Evaluating Femoral Anteversion: A Comparative Cadaveric Study

**DOI:** 10.3390/diagnostics12081820

**Published:** 2022-07-28

**Authors:** Kwang-Soon Song, Chang-Jin Yon, Yu-Ran Heo, Jae-Ho Lee, Seung-Bo Lee, Yeon-Kyoung Ko, Kyung-Jae Lee, Si-Wook Lee

**Affiliations:** 1Department of Orthopedic Surgery, Dongsan Medical Center, Keimyung University, 1035 Dalgubeol-daero, Dalseo-gu, Daegu 42601, Korea; skspos@gmail.com (K.-S.S.); poweryon@nate.com (C.-J.Y.); chzh5240@naver.com (Y.-R.H.); oslee@dsmc.or.kr (K.-J.L.); 2School of Medicine & Institute for Medical Science, Keimyung University, Daegu 42601, Korea; 3Department of Anatomy, Keimyung University College of Medicine, 1095, Dalgubeol-daero, Daegu 42601, Korea; anato82@dsmc.or.kr; 4Department of Medical Informatics, Keimyung University School of Medicine, Daegu 42601, Korea; koreateam23@gmail.com; 5MaiT Co., Ltd., Daegu 41931, Korea; koko4063@gmail.com

**Keywords:** femoral anteversion, computed tomography, axial oblique section

## Abstract

Twenty-five cadaveric adult femora’s anteversion angles were measured to develop a highly efficient and reproducible femoral anteversion measurement method using computed tomography (CT). Digital photography captured the proximal femur’s two reference lines, head-to-neck (H-N) and head-to-greater trochanter (H-G). Six reference lines (A/B in transverse section; C, axial oblique section; D/E, conventional 3D reconstruction; and M, volumetric 3D reconstruction) from CT scans were used. The posterior condylar line was used as a distal femoral reference. As measured with the H-N and H-G lines, the anteversion means were 10.43° and 19.50°, respectively. Gross anteversion measured with the H-G line had less interobserver bias (ICC; H-N = 0.956, H-G = 0.982). The 2D transverse and volumetric 3D CT sections’ B/M lines were consistent with the H-N line (*p*: B = 0.925, M = 0.122) and the 2D axial oblique section’s C line was consistent with the H-G line (*p* < 0.1). The D/E lines differed significantly from the actual gross images (*p* < 0.05). Among several CT scan femoral anteversion measurement methods, the novel anteversion angle measurement method using CT scans’ axial oblique section was approximated with actual gross femoral anteversion angle from the femoral head to the greater trochanter.

## 1. Introduction

Having an in-depth understanding of the morphological changes in femoral anteversion and accurately evaluating the condition are important in determining surgical deterioration in children with associated problems, such as in-toeing gait with torsional malalignment syndrome, developmental dysplasia of the hip [1], Legg–Calvé–Perthes disease, slipped capital femoral epiphysis, and cerebral palsy [2]. In addition, the morphological and anatomical features of the proximal femur are used in the preoperative planning for intramedullary nailing [3,4] and total hip arthroplasty.

By tradition, femoral anteversion measurement depends on the information obtained from a detailed physical examination using rotational profiles, including a trochanteric prominence angle test [2] and biplane radiography with specialized patient positioning or technique [5]. For this, computed tomography (CT) scans have become the most common imaging modality and the gold standard for clinical assessment due to their high accuracy and reliability [6,7]. Magnetic resonance imaging (MRI) and ultrasonography (USG) with new reference lines for measuring femoral anteversion were introduced as radiation-free techniques to reinforce CT scans. In contrast, other methods, such as 3D CT [8,9,10,11] and 3D biplanar radiography (3D-BR) [12,13,14], were also developed as new means of accessing the femoral version.

Even though 3D CT is considered a valuable diagnostic tool in evaluating bony femoral anteversion, it still presents several limitations, such as radiation exposure [15], a high level of inefficiency in terms of time and technical effort, and high cost. Moreover, severe deformity of the femoral head or neck-shaft angle or an inappropriate position makes it difficult to obtain 3D reconstructed images. However, the 2D CT technique exhibited excellent reliability and clinically acceptable accuracy [16].

The conventional transverse section of the CT scan has generally been recommended for use in measuring femoral anteversion. However, values were variable depending on the reference line used or the proximal aspect of the femur. The reformatted axial oblique section parallel to the femoral neck was also introduced for measuring femoral anteversion [11] based on the axial CT section and allowed for a more accurate anteversion assessment. This technique is less influenced by position and radiation, making it more similar to a conventional axial CT scan than a 3D reconstruction method [8,11,17]. Although CT has been widely used in evaluating the accuracy and reliability of anteversion measurement, there remains no established standard regarding its respective sections and reference lines. Moreover, no study has been designed to assess the consistency between actual anteversion and variable CT images employing the 3D reconstruction technique. Therefore, this study aims to identify the most reliable reference line for the anteversion of the proximal femur and the most valid and reproducible method for measuring femoral anteversion using CT scans.

## 2. Materials and Methods

Human cadaveric femoral bones were selected from the Department of Anatomy at Keimyung University, School of Medicine. Of the 30 cadaveric dry femoral bones collected, 5 bones were excluded due to their degenerated state—corrosion or breakage of the greater trochanter or femoral head area, leaving a total of 25 (7 right and 18 left) specimens included in this study. A consensus-building session to define each reference line was held with all the authors before measuring the anteversion angles. The observers are two experienced pediatric orthopedic surgeons and one anatomist. They estimated the anteversion from 75 measurements for each specimen that was obtained. The surgeons and the anatomist were blinded to the information about each specimen, including clinical records and results from other analyses differentiated by reference lines. Two distinct reference lines on digital photographs and four distinct reference lines on CT scans were used to measure the anteversion of the proximal femur. The distal reference line was defined as the posterior transverse connection line between the most prominent portion of the medial and lateral femoral condyles. All measurements were obtained using picture archiving and communication system (PACS) medical imaging technology (Marosis, DICOH version 5.0; INFINITT, Seoul, Korea).

### 2.1. Digital Photographic Technique

All cadaveric femurs were photographed by a digital camera placed on a cradle 1 m away while placed on a table that was always leveled. A sample was placed on a table with the lens of the camera proximally placed on the same collinear axis through the femoral head and bisecting both femoral condyles, representing the mechanical axis of the femur (Figure 1). The center of the head was determined by drawing a perfect circle around the femoral head and plotting its center. The most prominent portion of both distal femoral condyles was placed on the table horizontally and confirmed using an inclinometer. Measuring the angle of the distal reference line was deemed unnecessary.

#### 2.1.1. Head-Neck (H-N) Line

The H-N line is the connecting line between the center of the head and the center of the neck (Figure 2).

#### 2.1.2. Head-Great Trochanter (H-G) Line

The H-G line is the solid center line between the two dotted lines—one connecting the anterior border of the femoral head and the greater trochanter (GT) and the other connecting the posterior border of both femoral condyles (Figure 3).

### 2.2. Computed Tomography (CT) Scans

Each specimen was placed on an acrylic board with the most prominent portion of both the distal femoral condyles positioned horizontally. For proximal 2D CT slices, measuring the angle of the distal reference line could replace the acrylic board line (Figure 4). Femoral anteversion was measured as the angle formed by the proximal femur reference lines and the acrylic board. Five reference lines were used. The scan was carried out using the 64-slice multidetector CT scanner (SOMATOME Sensation 64, Siemens, Erlangen, Germany) with current at 40 mAs and voltage at 80 kVp level. Image pixel value was from 0 to 896. A 2 mm thick axial image was acquired through the specimen of dry cadaveric femora, followed by axial oblique multiplanar reformatting and 3D reconstruction using the imaging software (Syngo CT workplace IES, Siemens, Erlangen, Germany).

#### 2.2.1. A Line on a Conventional Transverse Section

In the coronal section, the transverse plane just below the head was selected. The anteversion angle was considered the connecting line between the center of the distal head and the center of the neck (Figure 5).

#### 2.2.2. B Line on a Conventional Transverse Section

Using coronal reformatted images (or topography) as a reference, the transverse section selected for measurement was through the femoral neck at the medial femoral head-neck junction. In the transverse section, the connecting line between the center of the partially visible distal femoral head and the center of the neck was considered the anteversion angle (Figure 6).

#### 2.2.3. C Line on an Axial Oblique Section

This novel reference line introduced by this study involves an oblique plane connecting the center of the head through the mid-portion of the basal neck and extending to the piriformis fossa area on the coronal section. On the axial oblique section, the middle line—situated between the line connecting the anterior border of the femoral head and the GT, and the one connecting the posterior border of the femoral head and the GT—was considered the anteversion angle (Figure 7).

#### 2.2.4. D Line on a 3D Conventional Reconstruction Section

In samples where the center of the neck could not be defined in craniocaudal view, the caudal to the cranial view of conventional 3D reconstruction was used instead. The anteversion angle is the angle between the two lines, one connecting the center of the femoral head and the mid-portion of the neck and the other connecting the posterior aspect of both femoral condyles (Figure 8).

#### 2.2.5. E Line on a 3D Conventional Reconstruction Section

This line was obtained from the cranial to the caudal view of conventional 3D reconstruction. The angle between two lines—one connecting the center of the femoral head with the mid-portion of the neck and the other connecting the posterior aspect of both femoral condyles—was considered the anteversion angle (Figure 9).

#### 2.2.6. M Line on the 3D Volumetric Reconstruction Section

This line was obtained from the cranial to the caudal 3D view of volumetric reconstruction. Mimics^®^ Innovation Suite (Materialise, Leuven, Belgium) software was used for the 3D model creation and measuring volumetric femoral anteversion. The angle between two lines, one connecting the volumetric center of the femoral head and neck and the other connecting the posterior aspect of both femoral condyles, was considered the anteversion angle (Figure 10).

### 2.3. Statistical Analysis

Statistical analyses were conducted using SPSS software (version 22.0E; IBM, Armonk, NY, USA). The interclass correlation coefficient (ICC) test was used to evaluate interobserver bias among H-N, H-G, A, B, C, and D lines. The ICC grade is estimated according to the study by Xiaoxia Han (2020). The consistency between digital photographs (H-N and H-G lines) and each CT scan (A, B, C, D, E, and M lines) was evaluated using the Wilcoxon test. Statistical significance was set at *p* < 0.05.

## 3. Results

The gross anteversion between the two lines had an excellent correlation coefficient, although the H-G line had slightly less interobserver bias than the H-N line (ICC; H-N line = 0.956, H-G line = 0.982). On the CT scans, none of the described methods showed interobserver bias (Table 1). In two samples (specimen nos. 6 and 18), the H-N line showed a negative value of anteversion despite the H-G line showing positive values for all three observers.

The mean values of the anteversion measurements for the H-N and H-G lines were 10.43° and 19.50°, respectively. Femoral anteversion in digital observations exhibited a significant difference (*p* < 0.01). The H-N line had a mean value smaller than that of the H-G line by 9.07°. The mean values of the anteversion measurements using the A, B, C, D, E, and M lines were 16.12°, 10.34°, 17.47°, 22.80°, 21.31°, and 8.59°, respectively. There were no significant differences in the H-N line measurements in the digital photograph except in the B line on the conventional transverse section (*p* > 0.05). Meanwhile, the H-G line in the digital photograph had no significant differences except in the C line on the novel axial oblique section of the 2D CT scan (*p* > 0.05). The H-N line had no significant difference compared to the B line on the 2D transverse section above the neck and M line in the volumetric 3D reconstructed image (*p* > 0.05; Table 2).

## 4. Discussion

Most torsional deformities in the lower extremities can be spontaneously improved in early childhood. However, in children and adolescents, the hip joint’s small size and anatomical variability according to developmental status, underlying disorders, and previous treatment may cause safety issues [18]. If torsional deformities of the lower extremities persist, clinical disability can introduce cosmetic problems in adulthood, from in-toeing gait to disabling hip and knee osteoarthritis. In eight-year-old children with rotational osteotomy and an abnormal anteversion angle of more than 50°, surgeons have reason to believe that femoral anteversion could be defined as the angle of the femoral neck relative to the femoral shaft when the femur is viewed along its long axis from above. However, there is still no standardized reference line or imaging modality of choice.

Direct anatomic measurement of the actual femoral anteversion angle, although ideal, has been impossible to carry out in a clinical setting. There are, however, many methods that are believed to be accurate and reliable for measuring femoral anteversion [2,8,9,11,13,14,16,19,20,21,22,23,24,25,26,27,28]. The computed tomography (CT) techniques using a single slice of the transverse section were generally advocated, describing the anteversion angle as the angle between the transcondylar line of the distal femur and the connecting line of the femoral head and neck. Furthermore, variable reference lines using single or multiple images were developed in different modalities [13]. In most studies, different reference lines were used per modality. For example, Lausten et al. [29] used the center head–neck line in CT but used the anterior head-trochanter line in USG, and their results indicated a poor correlation between CT and USG. Moreover, MRI, CT, and USG were compared through intrarater reliability, wherein MRI was the most reliable (r = 0.97/0.97), followed by CT (r = 0.99/0.96), and USG had slightly lower reliability (r = 0.88/0.88). The computed tomography (CT) has been believed to be the gold standard and showed more interobserver agreement than MRI in a study that uses the same axial oblique section [22].

The reference line containing the process for centralizing the femoral head, neck, and distal femoral condyles in multiple slices was developed to assess femoral anteversion more precisely. Although a projection of the proximal femur on the transverse plane was made to find the centroid of the femoral head and neck, the connecting line of those two centroid points could be considered femoral anteversion only if the femoral neck is perfectly cylindrical or if the neck-shaft angle is 90°. Moreover, it is overestimated compared to the traditional reference line that uses a single slice on the transverse section and is underestimated compared to the 3D reconstruction model. The anteversion angle had a significant difference at the level of the proximal femur. Meanwhile, the transverse section through the most proximal portion of the inferior neck (excluding the head) provided the most accurate estimate of the femoral neck axis in the study based on the 3D model [17]. In the current study, the anteversion at this level (B line) was consistent with the H-N line (*p* > 0.05). The B line was most closely approximated to the limited anteversion of the femoral neck. Notably, the mean value of the A line (16.12°) was not higher than the B line (10.34°).

Transverse scans pass through the neck obliquely. Moreover, the shape of the femoral neck is not cylindrical. Hence, it is evident that the anteversion angle should be changed according to its location and position. Measuring anteversion in the axial oblique section was first introduced by MRI [24] and CT scans soon after [11]. Moreover, reformatting an axial oblique section from the digital imaging and communication in medicine (DICOM) on the transverse section has advanced, preventing additional exposure to ionizing radiation. Using the CT technique, the axial oblique section also provided higher accuracy in measuring femoral anteversion independent of the femur position and allowed for a more accurate assessment of anteversion [11,17]. The axial oblique section showed the femoral head, neck, and GT in a single slice, and drawing the reference line was comparatively more straightforward in this study. The axial oblique section using the novel method reflects that the version from the center of the head to GT was consistent with the H-G line (*p* > 0.05). The C line most closely approximates the H-N line, which closely matches the conventional definition of anteversion, limiting the femoral inclination from the center of the head to the neck. Although both reference lines of the actual gross anteversion had good reliability, the H-G line had slightly less interobserver bias. As the three observers agreed, there is difficulty in pointing to the center of the neck using the H-N line because of the irregular shape of the proximal femur. Meanwhile, the H-G line was easy to draw, reflecting a valid and reproducible anteversion.

In the study that evaluated the proximal femoral geometry of 200 adult cadaveric femora using digital photographs, “femoral neck inclination” was developed as a new method to evaluate the true relationship between the femoral neck axis and the bicondylar plane. Additionally, the method, which has been widely used, could be significantly influenced by the neck-shaft angle, and does not represent a complex 3D anatomic relationship. Because of the geometrical relationship between the femoral neck and the shaft, the 90° view of the neck-shaft angle could represent the true neck inclination [30]. In another GT anatomical study, the varying shape of the GT was categorized into six kinds of morphological variations [3]. In addition, Chung et al. [2] reported the excellent validity and reliability of the trochanteric prominence angle test to measure femoral anteversion in cerebral palsy patients. To evaluate the precise extent of rotation between the proximal femur and bicondylar axis, anteversion, including the head to GT of the femur, is crucial in any clinical assessment. The axial oblique view was the only section representing the general axis from the center of the head to GT in a single slice image. The anterior head-trochanter line was reported as a reproducible method of measuring femoral anteversion, and the concept to reflect the general version of the proximal femur was added in this study. Specifically, the concept of the posterior head-trochanter line was included. By applying the midline of the anterior and posterior head-trochanter line as a reference, this novel method for measuring the femoral anteversion could replace other reference lines regardless of the variable and complex morphology of the proximal femur.

This study has some limitations. First, only adult cadaveric bones were used as a sample. Because many hip conditions occurring among children require clinical and research knowledge on the femur, studies should be able to use more child-centered materials and tackle more child-specific cases. This study was conducted using donated dry cadaveric adult femora, and the authors were blinded to the personal information of the samples. Second, the intra-observer bias in a single observer was not evaluated, which may have been necessary to improve reliability further and analyze intra-observer bias. In a 3D CT scan, reconstructed images cannot point to the center of the neck because of the complex structure of the neck-piriformis fossa and the metaphysis in the proximal femur. This study was unable to define the center of the neck in craniocaudal view in some of the cadaveric bones, requiring the use of caudal cranial projection. Despite previous studies supporting the accuracy of 3D CT [10,11,12,13,14,15,16,17], the D line exhibited a significant difference from the actual gross anteversion.

Regardless of these limitations, the study includes consistent test results for femoral anteversion between the actual gross photographs and CT techniques. The novel method used to define the reference line was unified into head-greater-trochanteric anteversion and axial oblique CT images. The double-blinded design and consensus-oriented process of determining the reference line offered evidence with reduced bias. Additional femur models that include the pediatric femora and 3D CT may have more predictive power in identifying the neck in a craniocaudal view.

## 5. Conclusions

This study demonstrated the most valid and reproducible method for measuring femoral anteversion using CT scans and represented how it identified the most reliable reference line for the anteversion of the proximal femur. The novel method for measuring the femoral anteversion angle using the axial-oblique section provided the best approximation of the actual gross femoral anteversion.

## Figures and Tables

**Figure 1 diagnostics-12-01820-f001:**
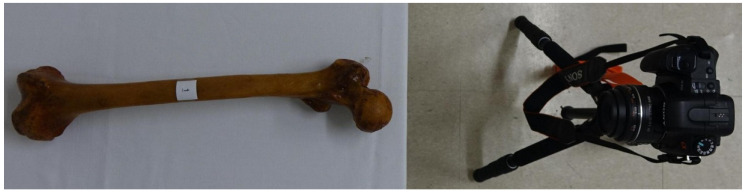
Method for measuring femoral anteversion using digital photography. A sample was placed on a table with the lens of the camera proximally placed on the same collinear axis through the femoral head and bisecting both femoral condyles to represent the mechanical axis of the femur.

**Figure 2 diagnostics-12-01820-f002:**
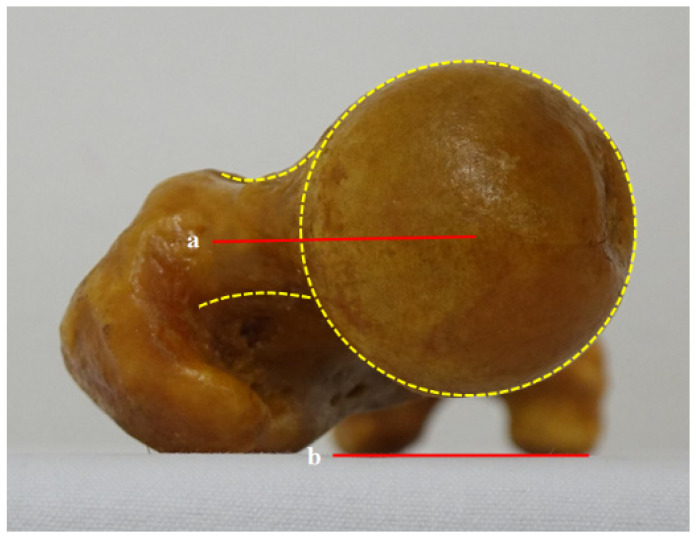
Femoral anteversion angle using the head-to-neck (H-N) line. (a) The connecting line between the center of the head and the center of the neck; (b) connecting line of the posterior border of both femoral condyles.

**Figure 3 diagnostics-12-01820-f003:**
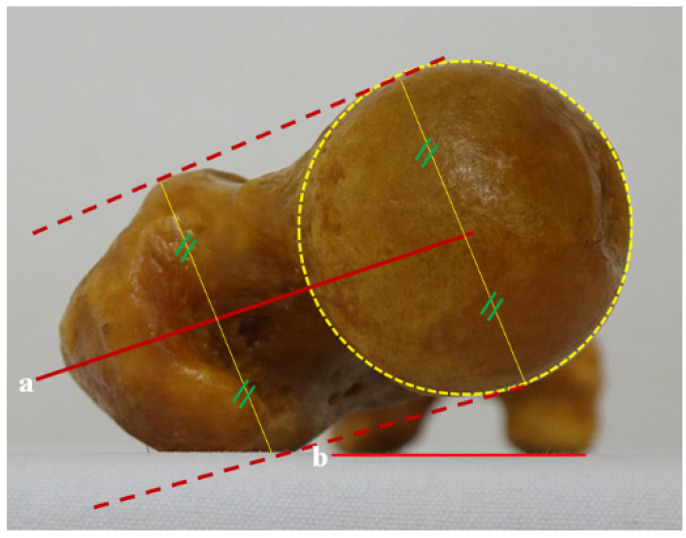
Femoral anteversion angle using the head-to-greater trochanter (H-G) line: (a) (H-G line) the solid center line between two dotted lines that connects the anterior and posterior borders of the femoral head and the greater trochanter; (b) the connecting line of the posterior border of both femoral condyles.

**Figure 4 diagnostics-12-01820-f004:**
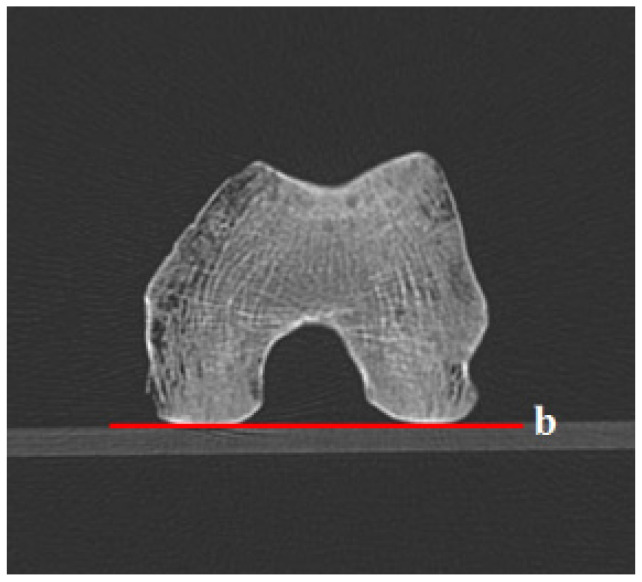
The distal reference line on the transverse section measures the femoral anteversion using 2D computed tomography (CT) scans. The b red line represents the connecting line of both femoral condyles’ most prominent posterior border.

**Figure 5 diagnostics-12-01820-f005:**
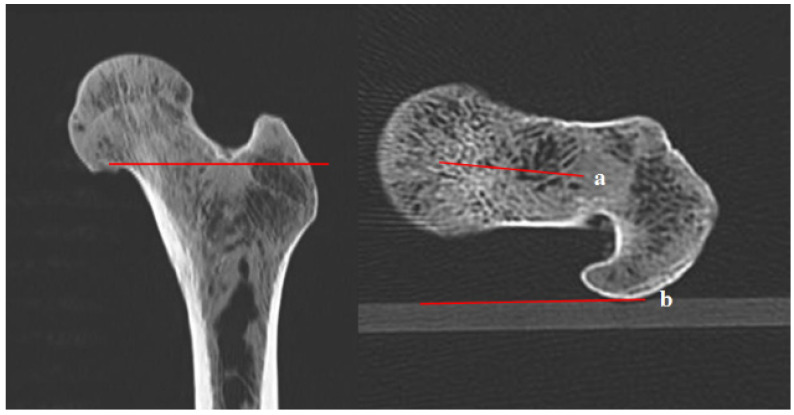
Left image: The femoral anteversion angle (angle between a and b) is calculated using the A line on the mid-femoral neck transverse section of a single transverse sectional image of the femoral neck just below the femoral neck (left image). Right image: (a) The connecting line between the center of the partially visible head and the center of the neck on the transverse section; (b) the connecting line of the posterior border of both femoral condyles.

**Figure 6 diagnostics-12-01820-f006:**
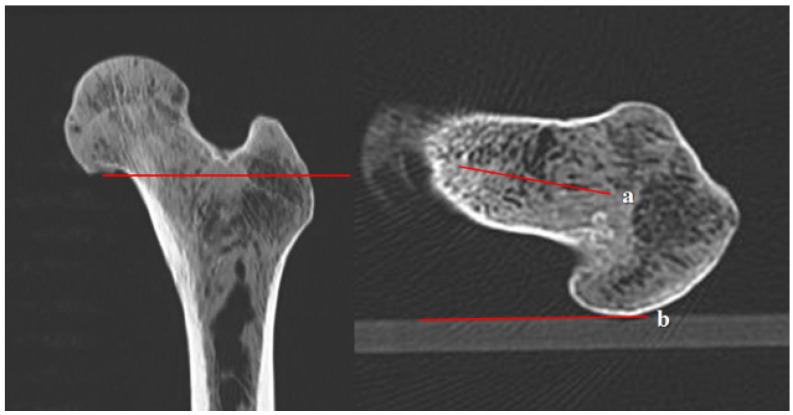
Left image: Femoral anteversion angle (angle between a and b) using the B line on the transverse section of a single transverse sectional image at the level of the mid-portion of the basal neck (left image). Right image: (a) the midline of the basal neck of the femur on the transverse section; (b) the connecting line of the posterior border of both femoral condyles.

**Figure 7 diagnostics-12-01820-f007:**
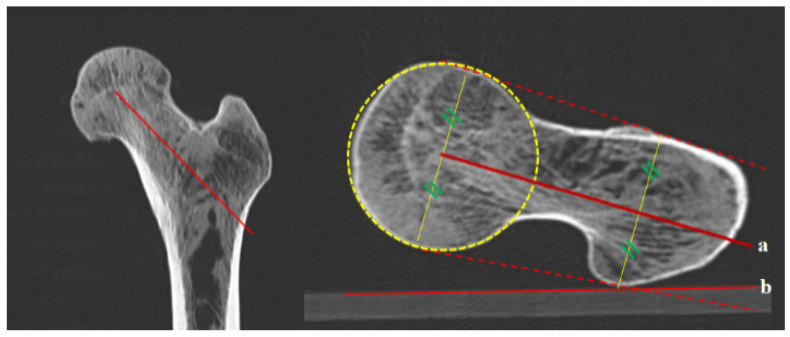
Femoral anteversion angle between a and b, using the C line on the axial oblique section. On the coronal section, the oblique plane that connects the center of the head and the mid-portion of the basal neck extends to the piriformis fossa. On the axial oblique section, (a) the middle solid line between the two dotted lines connects the anterior and posterior borders of the femoral head and the greater trochanter; (b) the connecting line of the posterior border of both femoral condyles.

**Figure 8 diagnostics-12-01820-f008:**
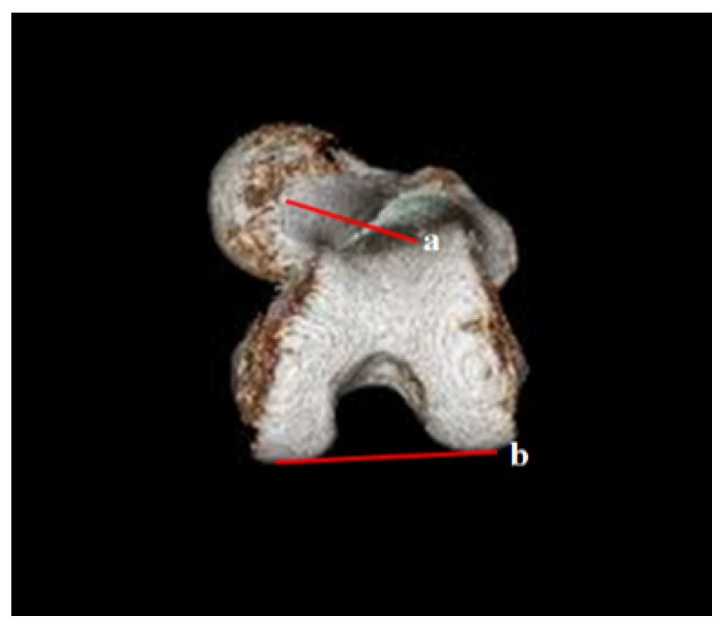
Femoral anteversion angle between a and b using the D line in the 3D reconstruction section of the caudal to cranial 3D reconstruction view: (a) the connecting line between the center of the femoral head and the mid-portion of the neck; (b) the connecting line between the posterior border of both femoral condyles.

**Figure 9 diagnostics-12-01820-f009:**
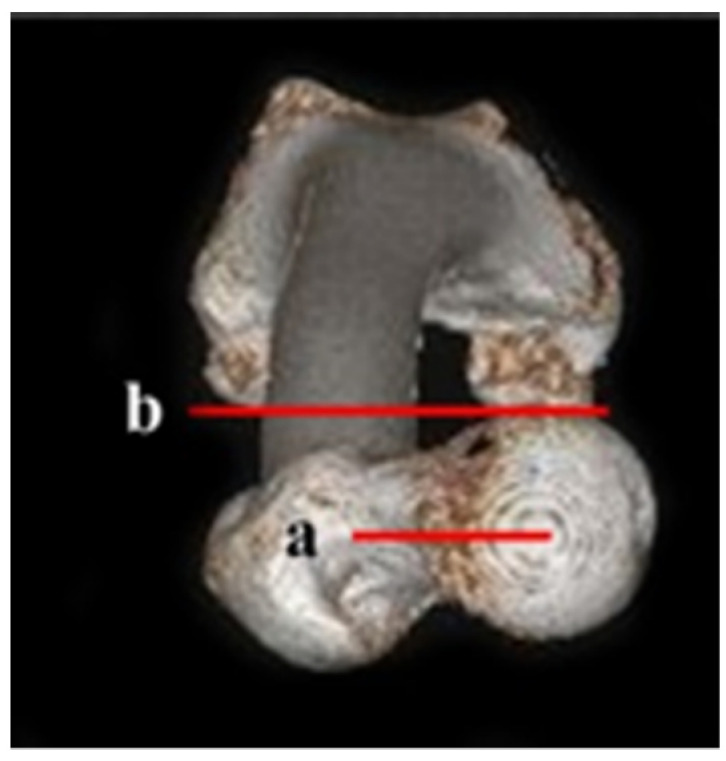
Specimen nos. 6 and 18. (a) Specimen no. 6, the H-N line showed the negative value of anteversion despite the H-G line showing a positive value for all three observers. (b) Specimen no. 18, the H-N line showed the negative value of anteversion despite the H-G line showing a positive value in all three observers.

**Figure 10 diagnostics-12-01820-f010:**
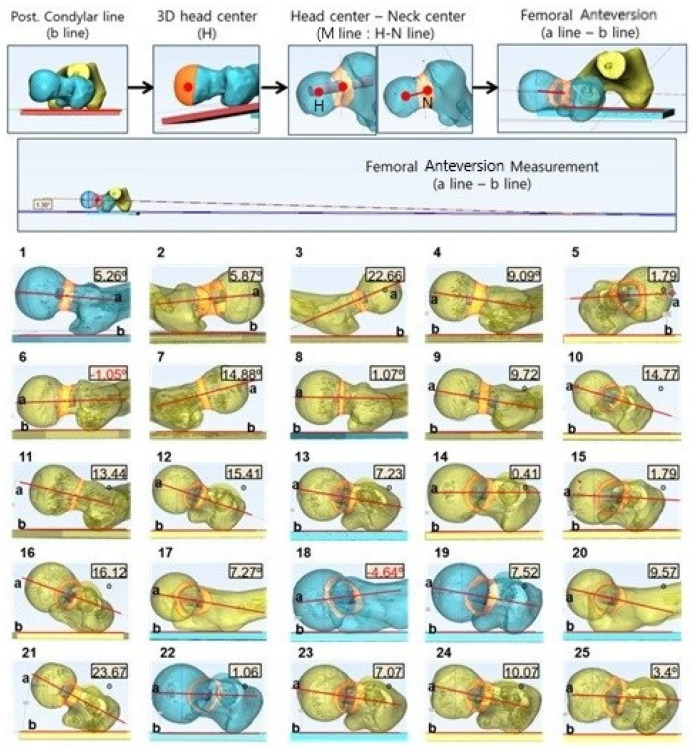
Femoral anteversion angle using the M line in 3D volumetric reconstruction was obtained from cranial to caudal in the 3D view of volumetric reconstruction. Mimics^®^ Innovation Suite (Materialise, Leuven, Belgium) software was used for creating 25 cadaveric femur 3D model and volumetric femoral anteversion measurements.

**Table 1 diagnostics-12-01820-t001:** Intraclass correlation coefficient by variable reference lines.

Line	ICC
H-N	0.956
H-G	0.982
A	0.994
B	0.996
C	0.986
D	0.965

A: anteversion on the transverse section just below the head; B: anteversion on the transverse section at the basal neck; C: anteversion on the axial oblique section using a novel method; D: anteversion on the 3D reconstruction section; H-N: head-neck line on the photograph; H-G: head-great trochanter line on the photograph; ICC: intraclass correlation coefficient.

**Table 2 diagnostics-12-01820-t002:** Consistency between digital photographic techniques and CT images.

Actual Gross Anteversion	CT Anteversion	*p*-Value
Reference Line (Mean ± SE)	Reference Line (Mean ± SE)	
H-N line (10.43 ± 1.12)	A line (16.12 ± 1.94)	0.000 *
	B line (10.34 ± 1.86)	0.925
	C line (17.47 ± 1.46)	0.000 *
	D line (22.80 ± 1.66)	0.000 *
	E line (21.31 ± 1.66)	0.000 *
	M line (8.59 ± 1.31)	0.122
H-G line (19.50 ± 1.00)	A line (16.12 ± 1.94)	0.021 *
	B line (10.34 ± 1.86)	0.000 *
	C line (17.47 ± 1.46)	0.065
	D line (22.80 ± 1.66)	0.010 *
	E line (21.31 ± 1.66)	0.021 *
	M line (8.59 ± 1.31)	0.000 *

SE: standard error; *: significant.

## Data Availability

The data presented in this study are available within the article.

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
