# Peer review of "Using the Axial Oblique View of Computed Tomography (CT) in Evaluating Femoral Anteversion: A Comparative Cadaveric Study"

_diagnostics, 2022, doi:10.3390/diagnostics12081820_

Round 1

Reviewer 1 Report

In this manuscript, the authors compared both digital photography and 2/3D CT images in measuring anteversion angle in femur. Overall, this manuscript provides a solid examination of different approaches with a clear outcome. However, some critical information is missing in the method section and some methodology descriptions are not clear. The manuscript should be revised regarding the following questions before publication.

Comments:

  1. Please indicate whether left or right  or both femurs were used.

  2. Imaging with cameras always faces the problem of image distortion and should consider image corrections with either grid paper or actually mechanical measurement unless proven small errors are introduced.

  3. CT imaging voltage, current and resolution missing. Image processing threshold (global or local thresholding).

  4. How is the coronal plane determined in Figure 5? Was it based on the whole femur or the midshaft?

Minor comments:

  1. Please check the grammar and rewrite the last sentence of the abstract, page 1, line 23-26.

  2. Demographic information of the samples (e.g., age, gender) are missing.

  3. The definition of H-G line is unclear. Figure 3, the two dotted lines are not always parallel to each other. Are the center of the dotted line used to create the center line? If so, please indicate the word “center” rather than using an ambiguous term of “in between”.

  4. More detailed information should be given to the ICC calculation due to the various number and classes of ICCs.

  5. It is not clear the meaning of  different color schemes(cyan and yellow color) of subpanels labeled with 1-25.

  6. The sentence in result section 2.2.1, line 128 and Figure 5 cation are not clear. Please indicate on the left image of Figure 5, the landmark being used on selecting the transverse plane just below the femur head.

Author Response

First, we appreciate your opinion.

Your mention is a lot of help for our paper.

Below is answer about your revision of our study.

Once again, thanks for your suggestion.

Comments:

Q1) Please indicate whether left or right or both femurs were used.

A1) Line72 : 7 right and 18 Left femur were used to evaluate the anteversion angle. 

Q2) Imaging with cameras always faces the problem of image distortion and should consider image corrections with either grid paper or actually mechanical measurement unless proven small errors are introduced.

A2) Line 87 : All cadaveric femurs were photographed by a digital camera placed on a cradle 1 meter away while placed on a table that was always leveled.

Q3) CT imaging voltage, current and resolution missing. Image processing threshold (global or local thresholding).

A3) Line 122 : 64-slice CT scanner (Sensation 64, Siemens Healthcare) with current at 40 mAs and voltage at 80kVp level. 2mm thickness slice was obtained and image pixel value was from 0 to 896.

Q4) How is the coronal plane determined in Figure 5? Was it based on the whole femur or the midshaft?

A4) Line 136 : The coronal plane was determined in midportion of femoral neck

Minor comments:

Please check the grammar and rewrite the last sentence of the abstract, page 1, line 23-26.

A) we have changed to Line23 “Among several CT scan femoral anteversion measurement method, novel anteversion angle measurement method using CT scans’ axial oblique section was approximated with actual gross femoral anteversion angle from the femoral head to the greater trochanter.”

Demographic information of the samples (e.g., age, gender) are missing.

Answer) Unfortunately, the donor's personal information cannot be shared, so it cannot be disclosed. However, it was confirmed that all of them used adult dry cadaveric femora.

The definition of H-G line is unclear. Figure 3, the two dotted lines are not always parallel to each other. Is the center of the dotted line used to create the center line? If so, please indicate the word “center” rather than using an ambiguous term of “in between”.

Answer) In figure 3, the H-G line is (a) line, it is displayed at line 111. Also, the two dotted lines is connected line between the margin of round shape of the femoral head and edge of the greater trochanter. So, the two lines are not parallel. And we have changed line 107, 112, according to the suggestion that indicate the word “center” rather than using “between”.

More detailed information should be given to the ICC calculation due to the various number and classes of ICCs.

Answer) The ICC grade is estimated according to previous study on by Xiaoxia Han 2020. We are explained at line 200.

It is not clear the meaning of different color schemes (cyan and yellow color) of subpanels labeled with 1-25.

Answer) in figure 10, the different color schemes are not meaning.

The sentence in result section 2.2.1, line 128 and Figure 5 cation are not clear. Please indicate on the left image of Figure 5, the landmark being used on selecting the transverse plane just below the femur head.

Answer) line 136 : The coronal plane was determined in midportion of femoral neck.

Reviewer 2 Report

General comments

This manuscript aims at identifying a reliable reference line for the anteversion of the proximal femur and a valid and reproducible method for measuring femoral anteversion using computed tomography scans. Developing methodologies different from radiation-emitting is always commendable (https://pubmed.ncbi.nlm.nih.gov/27421279/). In spite of some minor issues detailed below, authors manage to fulfill their aims sufficiently.

Minor comments

(line 74÷6) Please, re-phrase;

(l113 and elsewhere throughout MS) please, do not use acronyms in headings;

(l243 and elsewhere throughout MS) please, do not start sentences with acronyms;

(l331÷3) please, rephrase.

Author Response

First, we appreciate your opinion.

Your mention is a lot of help for our paper.

Below is answer about your revision of our study.

Once again, thanks for your suggestion.

Minor comments

(line 74÷6) Please, re-phrase;

Answer) we have changed line 74-76 to “They are estimated the anteversion from 75 measurements for each specimen were obtained.”

(l113 and elsewhere throughout MS) please, do not use acronyms in headings;

Answer) changed the word line 115.

(l243 and elsewhere throughout MS) please, do not start sentences with acronyms;

Answer) changed the word line 247 and 256.

(l331÷3) please, rephrase.

Answer) we have changed line 335-337 to “This study demonstrated the most valid and reproducible method for measuring femoral anteversion using CT scans and represented how it identified the most reliable reference line for the anteversion of the proximal femur.”

Reviewer 3 Report

This is a well-conducted study measuring the femoral anteversion angle by comparing the results of 2 differently calculated angles with two methods (digital photography and CT scans) in 25 cadaveric specimens. Specialized software was also used for detailed CT scans analysis. 

Results are also evaluated based on inter-observer reliability data, which denotes the authors' good understanding of measurement properties. ICCs were calculated. My suggestion to the authors is that ICCs are usually suggested to be complemented with SEM measurements, therefore please try to add these data for a more complete picture of measurement error limits (https://pubmed.ncbi.nlm.nih.gov/15705040/). 

Limitations of the study are appropriately addressed.

Author Response

First, we appreciate your opinion.

Your mention is a lot of help for our paper.

Below is answer about your revision of our study.

Once again, thanks for your suggestion.

comment : 

Results are also evaluated based on inter-observer reliability data, which denotes the authors' good understanding of measurement properties. ICCs were calculated. My suggestion to the authors is that ICCs are usually suggested to be complemented with SEM measurements, therefore please try to add these data for a more complete picture of measurement error limits (https://pubmed.ncbi.nlm.nih.gov/15705040/). 

Answer) The ICC grade is estimated according to previous study on by Xiaoxia Han 2020. We are explained at line 200.
